# Transformer Mechanisms Mimic Frontostriatal Gating Operations When Trained on Human Working Memory Tasks

## Abstract

The Transformer neural network architecture has seen success on a wide variety of tasks that appear to require *executive function* – the ability to represent, coordinate, and manage multiple subtasks. In cognitive neuroscience, executive function is thought to rely on sophisticated frontostriatal mechanisms for selective *gating*, which enable role-addressable updating– and later readout– of information to and from distinct "addresses" of memory, in the form of clusters of neurons. However, Transformer models have no such mechanisms intentionally built-in. It is thus an open question how Transformers solve such tasks, and whether the mechanisms that emerge to help them to do so resemble the gating mechanisms in the human brain. In this work, we analyze the mechanisms that emerge within a vanilla attention-only Transformer when trained on a task from computational cognitive neuroscience explicitly designed to place demands on working memory gating. We find that the self-attention mechanism within the Transformer develops input and output gating mechanisms, particularly when task demands require them. These gating mechanisms mirror those incorporated into earlier biologically-inspired architectures and mimic those in human studies. When learned effectively, these gating strategies support enhanced generalization and increase the models' effective capacity to store and access multiple items in memory. Despite not having memory limits, we also find that storing and accessing multiple items requires an efficient gating policy, resembling the constraints found in frontostriatal models. These results suggest opportunities for future research on computational similarities between modern AI architectures and models of the human brain.

## 1 Introduction

The Transformer architecture Vaswani et al. (2017) has recently become the dominant neural network model in artificial intelligence. Unlike some earlier AI architectures, which were inspired (albeit loosely) from human processing of language (Hochreiter, 1997) or vision (LeCun et al., 2015), Transformers have no mechanisms designed overtly to resemble the human brain. It thus remains an open question what, if any, similarities exist, and whether there are opportunities for theory or insights from AI and neuroscience to mutually inform one another (McGrath et al., 2024).

In this work, we focus specifically on working memory management via *gating mechanisms*. Gating mechanisms are responsible for multiple distinct aspects of working memory management, including determining which items to store and retrieve, when, and from where. In humans, there is strong evidence that such mechanisms are essential for tasks that require *executive function*, i.e., the ability to manage competing demands from multiple tasks, stimuli, and responses and to coordinate their execution (Frank & Badre, 2012; Badre & Frank, 2012; Chatham et al., 2014; Rac-Lubashevsky & Kessler, 2016; Rac-Lubashevsky & Frank, 2021). Although some Transformer variants have additional built-in structure for memory (Dai et al., 2019; Burtsev et al., 2020; Wang et al., 2019), the architectures which currently dominate modern AI systems are "vanilla" (Brown et al., 2020; Touvron et al., 2023), lacking specialized components to support this type of control.

We thus investigate whether, and under what conditions, such structure can emerge as a result of training. We train vanilla Transformer models on a task from cognitive neuroscience that was designed designed to investigate selective gating and working memory in humans (O'Reilly & Frank, 2006; Rac-Lubashevsky & Frank, 2021). We use recent techniques from *mechanistic interpretability* (Olah, 2022; Nanda & Bloom, 2022) to expose the mechanism that the Transformer uses to perform the task. We find that, as a result of training, the self-attention mechanism specializes in a way that resembles existing models of input-output gating (§4.1), but that these mechanisms only arise when the training task places specific demands on gating that mimics biological networks (§4.2). We further find that when such mechanisms do arise, they are predictive of better task performance and of generalization to changes in the input distribution and task demands (§4.3), improving effective working memory capacity (§4.4). Our findings highlight the importance of considering the emergent mechanisms that result from training in addition to the innate architectural mechanisms when drawing comparisons between AI systems and human cognitive processes.

## 2 BACKGROUND AND HYPOTHESES

There is strong evidence that working memory in human brains makes use of a *gating mechanism* to read, write, and maintain information required to carry out complex tasks (Rac-Lubashevsky & Kessler, 2016; Rac-Lubashevsky & Frank, 2021; Bhandari & Badre, 2018; Badre & Frank, 2012; Chatham et al., 2014). Sophisticated gating mechanisms of the type implemented in biological neural networks contain at least three important components. First, *input gating* controls whether or not given information is stored in memory, and if stored, determines the "address" (population of neurons) to which it should be written. Second, *output gating* determines when and what information to read out of memory to inform a subsequent decision, such as to produce a response to a task. Finally, working memory is *role addressable*, meaning that items can be bound to a learned task-dependent context (i.e. *role*) when stored and accessed in working memory. For example, in listening to a story for the first time, people will not just remember individual entities (*"Andrea"*, *"Chicago"*) but rather can associate them with specific roles such as lives_in(*"Andrea"*, *"Chicago"*).

In humans and other animals, these operations are supported by corticostriatal circuits in which isolated clusters of prefrontal neurons (or *stripes*) are used to represent distinct addresses in memory that can be updated or read out from via selective gating actions triggered by basal ganglia and thalamus (O'Reilly & Frank, 2006; Frank & Badre, 2012; Kriete et al., 2013; Calderon et al., 2022; Soni & Frank, 2024). These stripes can also serve as latent roles that condition how to interpret content within them. When learning effective gating policies, these models afford functions such as variable binding and indirection that support rapid generalization to new situations (O'Reilly & Frank, 2006; Frank & Badre, 2012; Collins & Frank, 2013; Kriete et al., 2013; Bhandari & Badre, 2018).

In principle, Transformers are good candidates for learning such gating behavior. Transformers' native self-attention mechanism consists of attention heads which are arguably functionally analogous to frontostriatal stripes. The decomposition of these heads into distinct keys, queries, and values (see Appendix A.1) means that the Transformer can in principle learn to differentiate reading and writing operations in a role- and context-dependent way across its multiple heads. However, whether Transformers will use their self-attention to implement such a mechanism is an open question, especially in cases when it is possible to fit the training data using more heuristic and less generalizable solutions.

We thus consider two hypotheses. First, we investigate whether, and under which data distributions, the Transformers use their attention heads to learn effective gating strategies. We find that the key vectors form addresses analogous to the PFC "stripes" (neural populations that support variable binding in memory). The learned key construction determines the address to store an item and is thus analogous to input gating. Conversely, the query vectors determine which addresses are accessed, and are thus analogous to output gating.

Second, we investigate whether adopting such strategies will facilitate rapid learning and generalization in working memory tasks the way it has been show to in humans. Specifically, human studies have shown that working memory capacity is not limited by the number of items one can maintain but rather by their effective gating strategies in frontostriatal circuits (Vogel et al., 2005; McNab & Klingberg, 2008; Baier et al., 2010). Theoretical work has shown that capacity limits in these circuits

do not stem from a limitation in the number of available neural populations, but rather result from a credit assignment problem that arises when learning to manage multiple items in memory (Soni & Frank, 2024; Todd et al., 2009). These limitations can thus be partially mitigated by learning to reuse effective gating policies (Soni & Frank, 2024). Because these limitations are computational rather than anatomical (i.e., not driven by the number of neurons/populations available), we hypothesize that they would also manifest in Transformers, even though they have no inherent memory demands at all (since all information is available in the context window).

# 3 EXPERIMENTAL DESIGN

## 3.1 TASKS

**Reference-Back 2 Task:**   We use a variant of a task from cognitive neuroscience known as the "reference-back 2" task (Rac-Lubashevsky & Frank, 2021). This is one among a number of task designs inspired by frontostriatal modeling work (O'Reilly & Frank, 2006; Soni & Frank, 2024) which requires selective updating and accessing of information in a role-addressable manner. In the reference-back paradigm, symbols are viewed one at a time with associated roles, and the participant must determine whether the current symbol is the same or different as that stored in memory for a given role. For example, a sequence might contain letters (role 1) and numbers (role 2), each of which occurs along side an update instruction which is either `Store` or `Ignore`. For each symbol in the sequence, the participant must do two things: 1) make a same/different judgment based on whether the current symbol matches the previously-stored symbol for that role, and 2) if the update instruction is `Store`, update the symbol associated with the associated role. See Figure 1 for an example. We create a modified text-based version of the reference-back 2 task. In our design, roles are denoted explicitly using special tokens (i.e., either `Reg0` or `Reg1` for the two-role version) indicating the role (or "register") to which the symbol should be bound. See Appendix A.2 for more details about our task implementation.

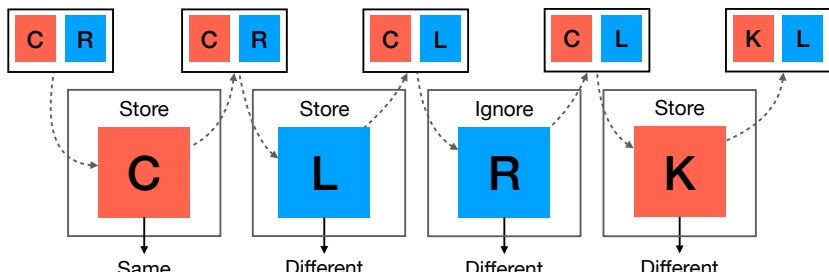

Figure 1: **Task** The reference-back-2 task requires making same vs. different judgments for each symbol in a sequence by comparing against a previously-shown symbol. See text for description of the task. In the above example, there are two roles, blue and red. The register state (shown along the top and connected by dotted lines) is assumed to be latent in the model; i.e., it is not provided as input.

**Split-Set Control:**   The computational advantages of role-addressable gating in frontostriatal networks (relative to other recurrent neural networks) are particularly evident when any symbol can be assigned to any register, so that the networks have to learn to assign them separable addresses (O'Reilly & Frank, 2006). To test the hypothesis that gating mechanisms only emerge in response to such task demands, we create a control condition in which the registers are associated with disjoint sets of symbols. In this task, it is possible to succeed without role-addressable gating, since symbols never need to be decoupled from their associated registers.

**Ignore-Integrated and Ignore-Separated Controls (a.k.a Split-Set Control):**   In the same vein, we include a further simplified task variant which is designed to require selective input gating but not output gating. This condition, along with the split-set control, have been previously used to show that the advantage of frontostriatal gating networks relative to other recurrent networks is largely reduced in such scenarios (O'Reilly & Frank, 2006).

Specifically, in the *ignore separated* condition (referred to as split-set control earlier), each register sees a disjoint set of symbols. Additionally, symbols falling under the `Ignore` update instruction are disjoint from those that are stored, meaning the model does not need to learn to differentiate `Store` and `Ignore` in a meaningful way. In contrast, in the *ignore integrated* condition, symbols under an `Ignore` instruction will be in distribution with the chosen register. In other words, the model must distinguish between a `Store` and `Ignore` instruction and thus learn input gating, but does not have to learn an output gating policy because both registers have mutually exclusive symbols.

### 3.2 MODELS

We train small, attention-only, decoder-only Transformer models from scratch on our task. Our models contain two decoder-only layers, each with two heads, and no multilayer perceptrons or layer normalization, followed by a linear "unembed" layer. The models are trained on 100k training data points for 60 epochs. See Appendix A.3 for additional details.

### 3.3 METRICS

**Performance on Refback2 Task:** We evaluated how well the model performs on the task on which it was trained using standard accuracy on the same vs. different prediction for each symbol in the sequence.

**Input and Output Gating:** The refback2 task is assumed to benefit from gating mechanisms, but success on the task is not in and of itself diagnostic of having learned the gating mechanisms. To develop an intrinsic measure of the input and output gating mechanisms, we use path-patching (Wang et al., 2022; Goldowsky-Dill et al., 2023), a generalization of causal mediation analysis (Pearl, 2001) that allows us to determine which components of a neural model (e.g., attention heads) work together in order to produce observed behavior on a task. Path patching involves designing a minimal pair of inputs, a "clean" input and a "corrupted" and then finding specific components of a model which fully account for the difference in the model's output between the two cases. By "patching in" only these components, the model can be made to behave as though it is seeing the corrupted input even when it is in fact seeing the clean input. See Appendix A.4 for more details.

We use path patching as a measure of input gating in the following way. For input gating, we design a minimal pair of inputs in which the corrupted copy includes `Ignore` in a place where the clean copy contained a `Store`, or vice-versa. We then use path patching to identify an incisive edit that can be made to the weights of the model in order to prevent the model from storing ("gating in") a given symbol. We report the accuracy of the input gating mechanism as the percentage of inputs on which this edit to the weights changes the models behavior in the expected way.

Analogously for output gating, we design minimal pairs which we expect to yield differences in an output gating mechanism, assuming one exists and is functioning correctly. Specifically, our clean and corrupted sequences differ in the register associated with one of the symbols, which should trigger a difference in what information is read ("gated out") in order to make a final same vs. difference judgment. Again, we report the accuracy of the output gating mechanism as the percentage of cases on which it is possible to make such an edit and produce the desired effect.

Our description of results in Section 4 elaborates on both this and the input gating metrics and their interpretation.

## 4 RESULTS

### 4.1 KEY AND QUERY VECTORS SPECIALIZE FOR INPUT AND OUTPUT GATING

We find that Transformers implement a role-addressable gating mechanism in which key vectors control input gating and query vectors control output gating (see Appendix A.1 for summary of key, query, value attention). Specifically, after training, the key for $\text{Sym}_i$ (e.g., at tokens 2, 6, 10, and 14 in Fig. 2a) represents the combination of the update instruction, register, and symbol for position $i$. A query's ability to address this position depends on whether the represented tuple contains a `Store`

Figure 2: **Path Patching Examples.** Model behavior across different path patching conditions. Attention is visualized as a shade of purple, with deeper shade corresponding to higher attention to that token. We create "corrupted" minimal pairs in which changing a token in the input (light blue) either changes the correct label at index 15 (examples b, c, e) or does not (d, f). We make small path-patching edits with the minimal pair to targeted network components (layer 1 keys for b, c, d, f; queries for e,f, see text). In all test examples, making the small patch successfully alters the model's prediction to align with the "corrupted" example, as expected.

or an `Ignore`. That is, key vectors representing an `Ignore` tuple receive very little attention (0.4% of layer 1 attention averaged over test set), whereas those representing a `Store` tuple receive the bulk of the attention (86.8%). We demonstrate the above narrative using path patching, described in Section 3.3), and further below. Figure 2 shows a summary of the path patching experiments and results. Our task is simple and does not contain noise, so in all cases, the intervention (i.e., patching to the keys or queries) results in a 100% change in the model prediction in the expected direction. Thus, for compactness, Figure 2 depicts the conditions but does not include quantitative results.

First, to investigate input gating, we create clean sequences sampled from our test set, and then corrupt these sequences by switching a `Store` within tuple $i$ to an `Ignore`. We path-patch only the key vectors of $i$. We expect, if the key controls input gating, that patching these key vectors should "block" attention to all of tuple $i$. An example attention pattern is in Fig. 2, examples a and b. We find that the model's attention shifts away from the tuple accordingly in 100% of patched instances. The presence of an `Ignore` or a `Store` within a tuple controls whether the key construction acts as an open input gate or a closed input gate.

Key construction also depends on the *role* of the represented content; within our task, that means whether $\text{Reg}_i$ is Register 0 or Register 1. When making a same/different prediction, key vectors representing a tuple that matches the target register receive most of the model's attention (92.5% of total attention), while those that do not match are not attended to (3.3% of total attention). Again, we use path-patching to determine that key construction encodes roles, this time perturbing the target register rather than the update instruction (see Fig. 2, row c). The model's attention shifts away accordingly across every example in the test set. Note that the stored tuple must be modified; if the same corruption is made earlier (as in row d), attention does not shift. This behavior shows that the gating within self-attention is *role-addressable*; the registers within the task function as roles, and are embedded within the key vectors as part of the representation.

Given that key vectors serve the role of addresses, query vectors in turn control which key vectors are accessed, through the final Q*K dot product in attention. Query construction thus performs the role of output gating within Transformers. The query composition controls which addressable Symbol i representations are attended to based on the identity of the target register. We again determine this through a set of path-patching experiments in which we perturb the target register (Fig. 2, row e).

We find that patching to the query vector in such cases indeed causes the attention to shift from the original stored tuple (74.1% of attention) to the stored tuple that matches the edited register, resulting in a corresponding change in the final same/different judgment. Editing aspects of the target tuple other than target register has minimal effect on the query construction. No edits to the query cause the model to attend to an `Ignore` tuple, further evidencing of output gating behavior– only content that has been made "addressable" can be accessed for a response. Furthermore, we

find that the target instruction and symbol do not factor into the query composition– changing them through path-patching to the query does not affect attention. This is notable because the model could employ other strategies for determining which tuples are eligible to be the stored tuple; e.g. attending to all symbols to match if any of them are the same as the target symbol.

## 4.2 EMERGENCE OF GATING POLICIES DEPENDS ON TASK DEMANDS

Under which conditions to gating computations arise? One possibility is that they are a trivial consequence of the Transformer's architecture. After all, gating is sometimes expressed as a simple multiplicative operation, as in LSTMs (Hochreiter, 1997). And indeed, the key, query, and value vectors underlying the attention heads are combined via matrix multiplications. However, this alone does not ensure that the networks learn effective, role-addressable gating policies. We thus investigate whether the training task demands induce such gating policies by running several control experiments (§3.1) which simplify the task, removing the role addressability of gating requirements. We thus hypothesized that since this task is much easier to learn, the Transformer will learn an overly memorized, brittle solution and will not develop effective input and output gating strategies.

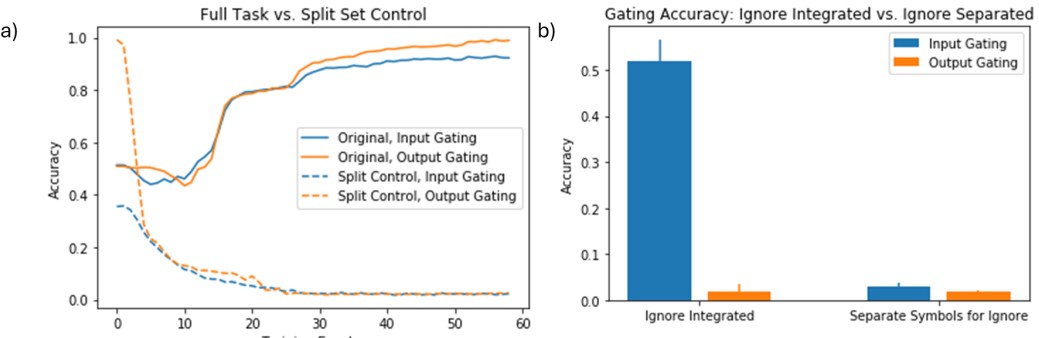

Figure 3: **Split Set Control** a) Ignore Symbols are not mutually exclusive with the registers. The models must learn to respond appropriately to store/ignore instructions (input gating). This control task separates the symbols shown to each register, reducing the need for the model to learn output gating. b) Both registers and the Ignore instruction have mutually exclusive symbols.

As shown in Figure 3, networks trained on the split set control task do not perform well at either the input or the output gating subtasks (see §3.3 for gating metrics). Moreover, models trained on the ignore-integrated task perform well at the Store vs Ignore input gating tasks, but not the output gating task, as expected, while models trained on the ignore-separated task perform poorly on both input and output gating (3b). These results strongly support the intuition that while Transformers have good inductive biases for learning role-addressable gating, it does not come "for free", and emerges only when demanded by the training task – the same contrast in demands that demonstrated advantages of frontostriatal gating (O'Reilly & Frank, 2006).

## 4.3 EMERGENCE OF GATING PREDICTS TASK PERFORMANCE

Is learning this gating policy useful for succeeding on the task? To answer this question, we first compare models with the same hyperparameters across different random seeds. We train 20 new models, each with a different random initialization, and measure both training loss and test set accuracy. 5 of the models succeed 100% of the time, and the other 15 models succeed between 94%-99.99% of the time, with a mean of 97.72% and a standard deviation of 2.03. We measure the intrinsic quality of the input and output gating subtasks using the path patching metrics described in Section 3.3.

Figure 4 shows the 5 runs that reach 100% accuracy on the test data as well as 5 randomly selected runs that do not. Two trends stand out. First, models which score a perfect test accuracy appear to succeed at the gating subtasks more readily than models which do not. Of the former, 3 of 5 models

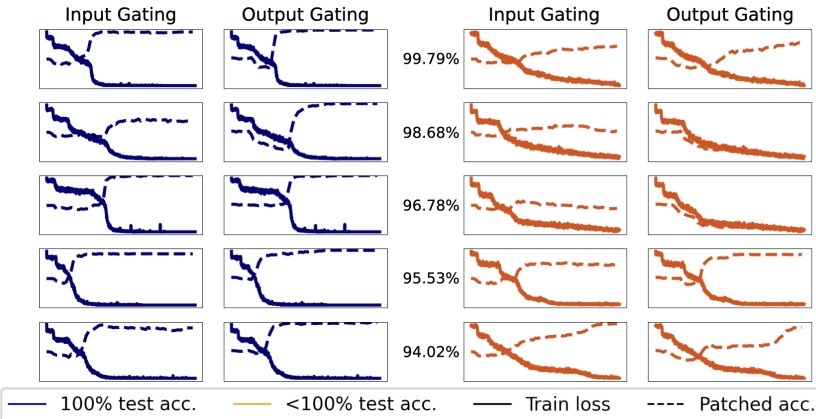

Figure 4: **Model Performance** over training on patching subtasks. Each graph contains an individual model's training loss (solid line) and subtask accuracy (dashed line, between 0 and 1) over time; the line's color corresponds to whether the model reaches 100% accuracy on the general test set.

reach 100% accuracy on both subtasks readily, plateauing less than halfway through training. In contrast, models that make errors in the test set also do not reach such immediate success at the subtasks (including the 10 not pictured in this graph); in fact, many categorically fail, scoring as low as 49% accuracy. These results do not indicate that this class of models' representations are useless for the task– they all score between 94% and 99.99%, well above chance performance.

The second trend is that many models across both classes have a sharp decline in training loss, which correlates with a similarly steep increase in accuracy on both subtasks. We interpret this phase transition as suddenly learning a gating mechanism. Models that do not exhibit phase transitions to the same degree take longer to fit the task, and do not reach high subtask accuracy.

While the above results suggest it is possible for a model to achieve fine accuracy in distribution without learning a gating policy, we hypothesize that models which do learn a more general gating strategy will better generalize to out of distribution examples. To assess this, we held out a subset of symbols from each register (randomized which symbol would be held out from which register). We tested these models on a challenge dataset which included examples of in-distribution pairings (register-symbol pairings that were seen in training) and out-of-distribution pairings (register-symbol pairings that were not seen during training). When holding out 5% of the symbols, the models learn a robust input and output gating strategy, and notably, perform on average at 99.7% for out-of-distribution symbol-register pairings. In contrast, the models trained on the split set controls do not learn robust gating strategies (despite performing perfectly at the trained task) and accordingly perform more poorly on the out of distribution examples: 77.2% (ignore integrated) and 88.6% (ignore separated).[1]

Note that the Split Set (Ignore Separated) task depends least on learning an effective gating strategy. These models learn the task very quickly and show no ability to perform on the gating subtask, and instead likely learn heuristic solutions. Interestingly, when tested on the challenge dataset (which includes examples of both in distribution and out of distribution register-symbol pairs), these models show a large disruption in their ability to perform on in-distribution sequences (80.8%). This insinuates that the strategy learned by these models is highly dependent on memorization and by adding in new elements, the strategy fails. The out-of- distribution accuracy is slightly higher than the in-distribution accuracy and future work could try to better understand the effects of perturbing a brittle model.

---

[1]One concern is that in the 5% held out, each register is trained on 49 symbols while in the split set (ignore integrated), each register is trained on 25 symbols. While all other parameters are held constant, this difference might be big enough to account for the large difference in generalization. To account for this, we ran another set of simulations holding out 60% of the symbols (each register is trained on 35 symbols). These models robustly learn a gating policy. Even with a drastic increase in the number of held out symbols from 5% to 60%, the out-of-distribution accuracy is still very high compared to either of the split set controls.

| Experiment | Out of Distribution | In Distribution |
|---|---|---|
| 5% Held Out | 99.7% | 99.8% |
| 60% Held Out | 95.3% | 99.3% |
| Split Set (Ignore Integrated) | 77.2% | 94.6% |
| Split Set (Ignore Separated) | 88.6% | 80.8% |

Table 1: **Accuracy for In Distribution and Out of Distribution Symbol-Register Pairings in Challenge Dataset** Experiments and their associated performance on the challenge data set. The challenge dataset includes a mix of in-distribution and out-of-distribution symbol-register pairings. This table breaks down the accuracy based on if the symbol-register pairing was in the training (In-Distribution) or not seen during training (out-of-Distribution).

### 4.4 GATING POLICY TRANSFERS TO INCREASED TASK DEMANDS

Thus far, the experiments have focused on tasks with two registers, mimicking that used in the reference back-2 task (Rac-Lubashevsky & Frank, 2021). However, human working memory has a capacity of about 3-4 items (Cowan, 2008), albeit with vigorous debates questioning whether this limit is discrete or continuous (Zhang & Luck, 2008; Wei et al., 2012; Luck & Vogel, 2013). More recent models and data suggest a "chunking" hybrid between the two, whereby multiple memoranda can compete for shared continuous resources within discrete slots (Nassar et al., 2018; Soni & Frank, 2024). Chunking increases *effective* capacity, allowing more items to be remembered at the cost of precision of some of the items. Notably, in frontostriatal gating models, when the number of registers to manage was larger than two, networks given limited memory allocation but with chunking capabilities performed better than those that were allocated as many PFC populations as items to store (Soni & Frank, 2024). The reason for this seeming paradox is *credit assignment*: as the number of PFC populations (stripes) increases, the gating management problem becomes more challenging – the network has to learn to route each item to distinct populations and to also learn to read out from the corresponding population for a given probe. Moreover, these learning problems are interdependent.

These limitations are computational rather than anatomical and stem from the learning process. We hypothesize that they would also manifest in Transformers, even though Transformers have no inherent memory demands (since all information is available in the context window). Specifically because of the flexibility and expressivity of Transformers, we predicted that by increasing task demand, the network would learn the task, but struggle to do so. We further predict that models that learn mechanistic solutions will generalize better to new tasks.

To test this hypothesis, we increased the number of registers from 2 to 3. First, we confirmed that asymptotic accuracy dropped to 95.4%. This result is qualitatively the same even when training for twice the number of epochs. This is non-trivial given that we are just adding one register. We predicted that the degree to which the transformer solves the task is related to heuristics and memorization rather than gating in these cases. We predicted that if we first pretrained the model to effectively manage two registers, the network will be able to scaffold the learned gating strategy to learn the three register task more robustly. We further predicted that this pretraining would only be useful if pretraining encouraged gating strategies. Our results in Figure 5 support both of these conclusions. Not only do pretrained networks on the original reference back-2 task exhibit higher gating sub-task accuracy 5, but networks that were pretrained with the original two register task showed very rapid learning in the three register task in the first few epochs. Moreover, this pretraining was far less effective when it did not encourage gating (the split symbol control from above).

## 5 SUMMARY AND DISCUSSION

In this work, we investigate Transformer models for emergence of a learned *gating mechanism*; a network component performing role-addressable gating, similar to that in working memory of humans. We observe that the model learns a gating policy and find that task performance is correlated with gating ability. Our results show how learning gating mechanisms is one way Transformers can excel at tasks that require executive function.

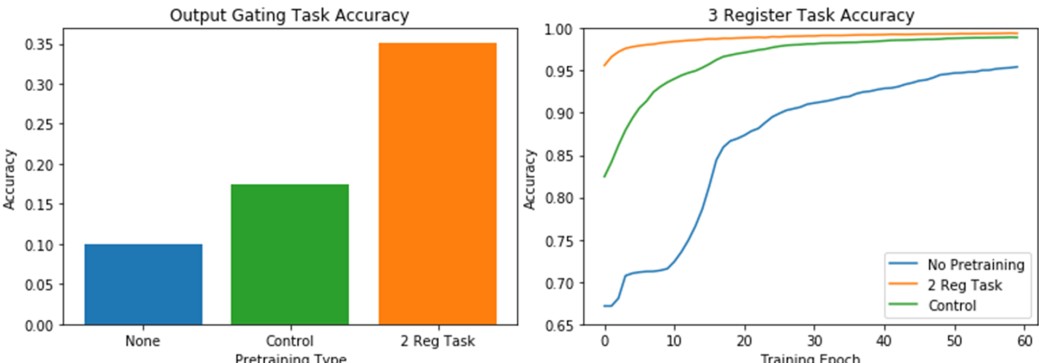

Figure 5: **3 Register Task and Pretraining** a)At the end of training on the 3 Register Task, models with 2 Register Task pretraining perform the best on output gating tasks, insinuating learning of a mechanistic solution. b) Accuracy curves through training (after the pretraining) show a stark difference between pretraining on the normal 2 register task and the control task. Models pretrained on the 2 register task, generalize quickly (first few epochs) and with high accuracy (above 95%) on this new task. At the end of training, no pretraining models are below 95% accuracy - which is below the accuracy that 2 register task pretraining models showed at the beginning of training. Control pretraining models show some generalization (due to learning of basic elements e.g. symbols) but do not generalize to the same degree. Models pretrained with the 2 register tasks, which should encourage a mechanistic solution, perform well at the new 3 register task, showing a superior ability to generalize. 22 random seeds were run for each experimental condition and the results here an average of those models. See Appendix A.5) for comparison on input and output gating task accuracies.

The Transformer models are capable of making use of key composition for input gating and query composition for output gating on the task. We find that making precise corruptions to specific architectural elements of the network causes the model's prediction to change from *Same* to *Different* or vice versa, indicating that those components are causally responsible for the gating mechanism. The architectural biases of attention within the vanilla Transformer model lend themselves well to representing role-addressable content. The learnable nature of keys, queries, and values allows the model to learn to create internal representations. These representations can be learned to signify roles and addresses, mimicking the variable binding and input / output gating mechanisms in biological neural networks (O'Reilly & Frank, 2006; Frank & Badre, 2012; Collins & Frank, 2013; Kriete et al., 2013).

When we trained more models on the task, we found that the models which perform best on the task correlate with the markers of gating we observed in our circuit analysis, and that the learning trajectory shows a steep decrease in training loss and a steep rise in patched subtask accuracy simultaneously, suggesting that the model has learned a gating policy at that time. Both findings are analogous to those of Frank & Badre (2012), in which they find that networks which learned a hierarchical gating policy performed better at a hierarchical learning task, and humans that learn this policy also show a sharp decrease in loss when they discover it. There is still more work to be done to better understand the models that don't learn the gating policy: what types of solutions do they learn? How do we push models towards a mechanistic solution by hyperparameter tuning? Understanding grokking phase planes in a similar manner as Liu et al. could be informative.

Nevertheless, we show that a critical factor controlling the learned gating policy is the task demands. We found that experimental conditions in which biological networks exhibit gating advantages were similarly needed to give rise to learned gating policies in Transformers. Conversely, when the task places less demands on role-addressable gating (our split symbol control conditions), the models 1) did not learn gating policies, 2) were less able to generalize to out-of-distribution pairs, and 3) were less able to rapidly acquire tasks with higher gating demands (three registers).

We show that models that learn and solve the tasks using gating mechanisms are better at generalization. Further we show how learning brittle and memorized solutions causes some models to falter at

even in-distribution pairings when they are mixed with out-of-distribution examples. These results can be used to inform larger models that are needed for better and faster generalization. Future work can characterize what kinds of mistakes are common within the mechanistic and heuristic models to further characterize these models and promote better generalizability.

Ultimately, finding connections between emergent behavior of Transformer models and human working memory serves to benefit both computational cognitive neuroscience and artificial intelligence. Although Transformer models themselves are limited in their biological plausibility, in this setting they learned behavior mimicking the functionality of working memory, and their application within computational models of the brain should be further explored. From the perspective of artificial intelligence, understanding the strengths and limitations of Transformer models on executive function tasks may inform model analysis across the many diverse settings in which these models are applied.

## 6 LIMITATIONS

While this paper draws parallels between the Transformer architecture and the brain, it is important to emphasize that there are some significant differences between how Transformers and humans solve tasks. In particular, because Transformers can attend to any part of the sequence when creating a representation, they are not limited by memory constraints. Transformers can solve tasks that would push the limits of human working memory (but it should be noted that Transformers require disproportionately large amounts of training data to do so). Nevertheless, we hypothesized that Transformers might still learn effective gating policies that mimic those in frontostriatal networks. Moreover, as briefly reviewed in the introduction, working memory capacity in humans and biological computational models is not limited primarily by the memory demand *per se*, but rather by the difficulty of the credit assignment problem for learning how to manage role-addressable storage and access of multiple items in memory. A fundamental bridge between Transformers and other models of WM (and humans themselves) would be if Transformers also needed to overcome the credit assignment learning problem, despite an unlimited memory capacity.

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

## A  APPENDIX

### A.1  SELF-ATTENTION IN TRANSFORMERS

Transformers are powerful language models which create contextualized representations of the input sequences. They learn to predict the next token one at a time using an "attention" mechanism that scans other tokens in the sequence for relevant information (Bahdanau et al., 2014). A common practice (that is used here) is to mask any future tokens and so predictions and representations must be made by the current token and any previously seen tokens. These models are able to learn and represent complex sequence modelling tasks.

For a given prediction, Transformer attention generates three separate vectors for each token in the sequence: a query, key, and value ($q$, $k$, $v$). The key vector is a set of tokens that the model has learned are most relevant to the token at hand. The query vector scans the tokens in the key vectors and calculates how much the current prediction should "attend" to each of those tokens. Then, the value vectors at those positions are multiplied by the corresponding weights, summed up, and added to the next representation: for token $i$ at layer $j$, the contextual representation is $\sum_k q_i^j \cdot k_k^j * v_k^j$. Thus, the next token prediction includes earlier sequential information by combining the value vectors from previous tokens. In other words, Transformer attention can be viewed as a read/write mechanism: for a given token, the queries and keys dictate which tokens to read from, the values are the content that is read (proportional to the attention calculated by the keys and queries), and the summed content is written to a new representation at the given token. As we shall see below, the comparison to role-addressable input and output gating operations is evident. The key vectors form addresses analogous to the PFC "stripes" (neural populations that support variable binding in memory). The learned key construction determines the address to store an item and is thus analogous to input gating. Conversely, the query vectors determine which addresses are accessed, and are thus analogous to output gating.

## A.2 TEXTUAL REFERENCE-BACK-2 TASK

The *textual reference-back task* requires making same/different judgments between incoming symbols assigned to a particular "register" in memory, with respect to those seen previously and linked to those same registers. Like the original tasks, the textual reference-back task is sequential, and requires independent updating and maintenance of two memory registers, each containing one of $S$ arbitrary *symbols* at a time. At the beginning of each sequence, each register is initialized individually to one $s \in S$ (the pool of symbols is shared between registers, which was shown to more substantively tax gating mechanisms in (**?**).) Each sequence is composed of $L$ tuples, each containing register address $\text{Reg}_i$, symbol $\text{Sym}_i$, same/different label $\text{Ans}_i$, and update instruction $\text{Ins}_i$. For a tuple $i \in L$, the **answer** $\text{Ans}_i$ is a binary value that is either Same if symbol $\text{Sym}_i$ is currently stored in the register with address $\text{Reg}_i$, or Different otherwise. The **update instruction** $\text{Ins}_i$ also takes one of two values (`ignore` or `store`), evenly distributed. If the instruction is `ignore`, then the model still needs to make the same/different determination with respect to the stored reference, but the new symbol should not update the register content (i.e., the reference remains unperturbed). If it is `store`, then from that point on in the sequence, $\text{Sym}_i$ is stored in the register with address $\text{Reg}_i$ until otherwise updated. An example is shown in Fig. 1.

We implement each reference-back task example in our data as a single sequence, and measure models' ability to predict Same versus Different for each $\text{Ans}_i$. Each sequence has 10 same/different answers, and we generate 100,000 train, 1,000 validation, and 1,000 held-out test sequences.

The class balance of `same` to `different` answer labels in the train/test datasets is roughly 1:2, making a "maximum class" heuristic solution 0.66 accuracy, 0.33 precision, and 0.5 recall. We test several other heuristics, the strongest of which is predicting `same` if another tuple including `Store` and the target register and target symbol exists in the sequence, which scores 0.80 accuracy, 0.82 precision, and 0.85 recall.

## A.3 MODELS AND TRAINING

We train small, attention-only Transformer models from scratch on our task. Our models contain two decoder-only layers, each with two heads, and no multilayer perceptrons or layer normalization, followed by a linear "unembed" layer to project the output of the last decoder into the space of the entire vocabulary at each timestep In practice, only 'same' and 'different' are ever predicted. Our network uses absolute positional embeddings (Vaswani et al., 2017). The vocabulary contains all possible tokens, represented individually with embedding size $E$. Models are trained to predict the next token with the language modelling objective: if the model is predicting $\text{Ans}_c$, it will have access to all ($\text{Ins}_i$, $\text{Reg}_i$, $\text{Sym}_i$, $\text{Ans}_i$) tuples where $i < c$, as well as $\text{Ins}_c$, $\text{Reg}_c$, and $\text{Sym}_c$. However, the models only receive loss at positions where a same/different token must be predicted (this is similar to the reward function applied in frontostriatal gating networks; (O'Reilly & Frank, 2006; Soni & Frank, 2024)). Furthermore, each layer gets a causal attention mask– when constructing each token representation, it cannot look ahead at tokens further down the sequence.

The models are trained over 60 epochs of the 100k training data points, learning from 6 million examples in total. Models are evaluated on their accuracy (whether the correct $\text{Ans}_i$ is predicted for each tuple $i$), measured in precision and recall, as well as the `same` versus `different` token logit difference.

## A.4 PATH PATCHING

Path-patching involves making a incisive edit to the representations of a trained model and observing how the model's behavior is affected, allowing one to infer the computations implemented within individual attentional heads (see Fig. 6). Path-patching requires a minimal pair of examples: the "clean" example and the "corrupted" example, in which one token from the clean example is changed, as well as the correct label. Given representations from the model for both the clean and the corrupted examples (the blue and orange components in the figure), we can chose a specific component anywhere in the model (referred to as the "sender"), and replace the clean representation with the corrupted one at that specific component. These embeddings will be received by the next layer ("receiver"), thereby "patching" the path. From there, the patched model will compute the new prediction.

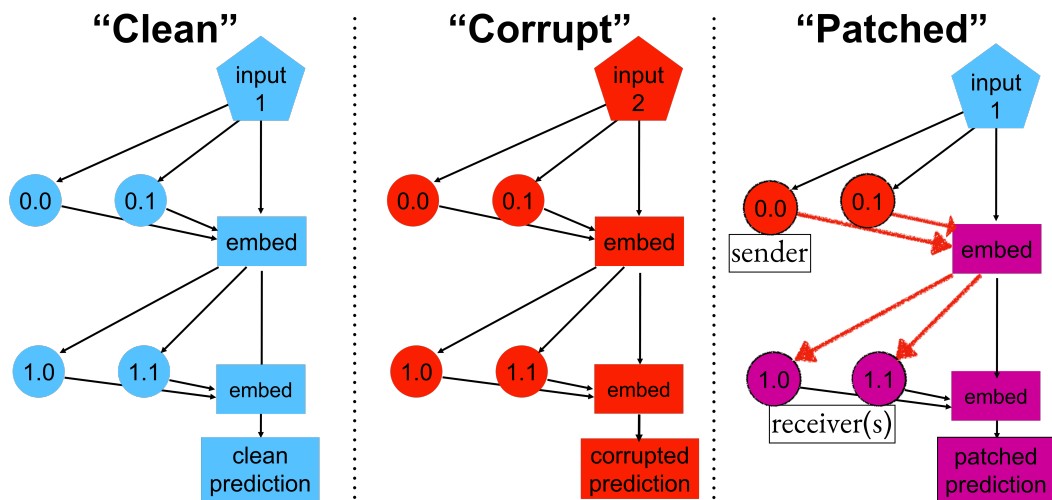

Figure 6: **Graphical Path Patching** Graphical diagram of the path-patching process. Attention heads are represented as circles (layer,head index), and contextual representations of each token (as well as the next token prediction) are represented as rectangles.

In the figure, we send from layer 0, head 0 and 1 to layer 1, both heads 0 and 1. All clean representations that are not along this path are not modified and are unaffected by the patch. The model then recomputes all representations after the receiver (the "patched" representations), and arrives at a new prediction. If the model output matches the corrupt prediction rather than the clean one, that prediction is causally dependent on the path from sender to receiver. See (Wang et al., 2022) and (Goldowsky-Dill et al., 2023) for a more comprehensive review of path-patching methods.

We perform a small hyperparameter search and select a model that reaches 100% accuracy on the held-out test data for further analysis. We determine the circuit that the model uses through an array of path-patching experiments with a simple minimal pairs paradigm. Our "sender" within path-patching is always both attention heads at layer 0, and our "receiver" is always both attention heads at layer 1.

We first establish that a Transformer model is able to succeed on the reference-back-2 task.

At layer 0, the model learns to condense the task-critical information from each tuple into one embedding, at the position for $\text{Sym}_i{}^2$. At this layer, the model pays 85.8% of total attention to the task-critical information to that tuple, and just 14.2% of attention to other tuples.

At layer 1, the attention heads learn to attend to the $\text{Sym}_i$ key vector representing the tuple where information was last stored in the target register. The heads pay 70.2% of total attention to this tuple (the "stored" tuple), and only 29.8% of attention to all other tokens. This behavior is tied to the target register matching the register in the stored tuple, which is analogous to gating of the relevant role-addressable PFC stripe (O'Reilly & Frank, 2006; Soni & Frank, 2024). We focus our analysis on the Layer 1 representations which exhibit this learned gating policy, shown in Fig. 2.

A.5 Input vs. Ouput Gating Accuracies, Pretraining

---

[2]Redundantly, the model does the same at the position for $\text{Ans}_i$. Through additional experimentation, we determine that this is a quirk of Transformer learning, and does not impact our analysis.

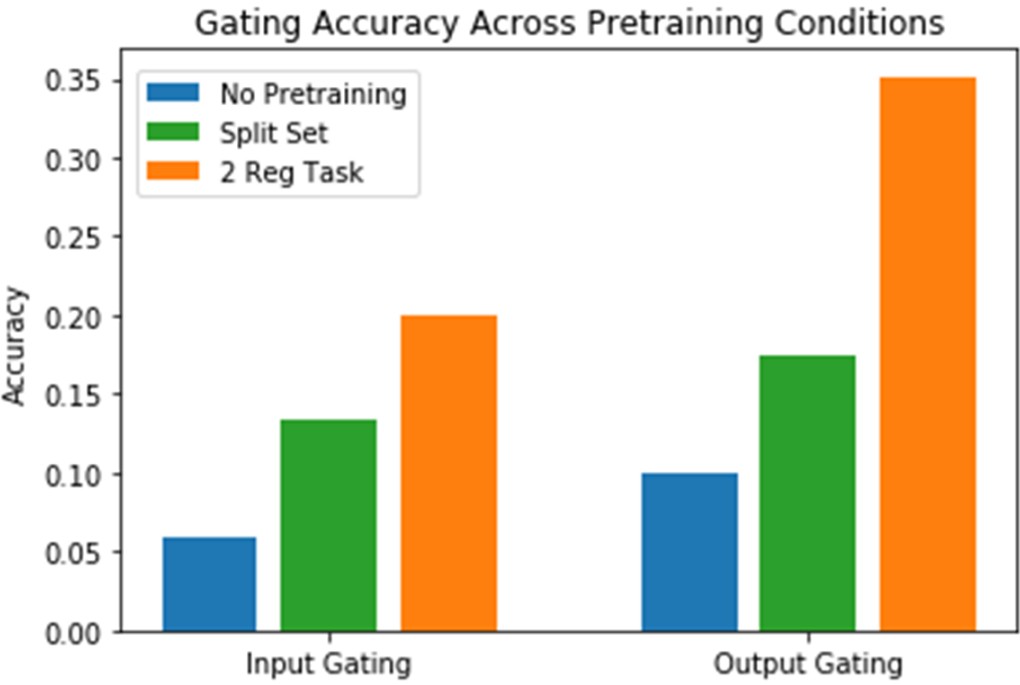

Figure 7: **Input and Output Gating Task Accuracy** Pretraining on the 2 register task leads to the highest accuracy on input and output gating subtasks - insinuating that these models learn the most mechansitic solutions. In general the input gating accuracy is lower, indidcating a differential role for each gating. There is human experimental evidence to suggest that output gating is harder to learn and is more crucial for better performance

