# OpenReview forum: "Transformer Mechanisms Mimic Frontostriatal Gating Operations When Trained on Human Working Memory Tasks"
_ICLR.cc/2025/Conference — Submitted to ICLR 2025_

### Official Review · Reviewer_2znX · 2024-10-30

**Soundness:** 3
**Presentation:** 1
**Contribution:** 1
**Rating:** 5
**Confidence:** 4

**Summary:**

This manuscript aims to understand the conditions by which interpretable gating mechanisms emerge in self-attention using attention-only transformers. The authors train simple, attention-only transformers on a simple reference-back-2 task. Using path-patching approach to interpret transformer mechanisms, the authors characterize what the query and key vectors do/implement during the reference-back-2 task and related control tasks. Overall, they find that the query vector accesses information from specific tokens within the context window (output gating), whereas the key vector provides the address to store an item to gate information (input gating).

**Strengths:**

Altogether, the results form an intuitive and helpful story to improve the understanding of the transformer’s self-attention mechanism. The combination of the simplicity of the task, and the path-patching approach on the attention-only transformer are instructive tools for understanding.

**Weaknesses:**

Much of the paper focuses on trying to understand the mechanisms of self-attention. Besides a general concern that I find the results somewhat unsurprising, I have three other weaknesses I’d like to highlight.

1. Novelty. There are a number of papers in mechanistic interpretability that aim to study how transformer mechanisms (including self-attention) influence output/performance via path-patching, e.g., Meng et al., 2023a, 2023b; Olsson et al., 2022, Li et al., 2023 etc. However, most the background in the paper references neuroscience theories (which is fine, but perhaps misleading as to existing work in mechanistic interpretability). A more thorough contextualization of mechanistic interpretability papers would be beneficial.

2. Though the paper is geared towards understanding self-attention mechanism through experimentation, I found the manuscript to be difficult to understand. For example, I found the presentation of the task (both text description and figure 1) to be quite confusing. Only when I reviewed the text and figure from another paper (Rac-Lubashevsky & Frank, 2021) did I understand the task. It shouldn’t be necessary to review a prior paper to understand the experiment. I found figure 2 to be also quite dense and confusing. (I have previously found the figure presentation in Meng et al., 2022, Fig 1 to be quite instructive.) Finally, many of the analyses (e.g., Queries gate outputs; keys gate inputs) are only reported in text. Would it be possible to include a visual figure/understanding that is intuitive as to what the query and key vectors are actually computing?

3. The pretraining result is generally unsurprising (fig 5), and is consistent with prior results in curriculum learning, which is not mentioned.
This is perhaps more of a comment than an explicit suggestion, and something for the authors to consider: In some ways, I think contextualizing the work using the background of ‘frontostriatal gating’ mechanisms may be confusing to some readers. I understand that trying to demonstrate this biological/neural mechanism in transformers may have been the motivation to the authors, yet to a reader who is unfamiliar with neuroscience, this may not add much. The explanation of gating mechanisms via path-patching should in theory computationally sufficient, and doesn’t require reference to neuroscience (which can often be obscure).

Meng, Kevin, David Bau, Alex Andonian, and Yonatan Belinkov. “Locating and Editing Factual Associations in GPT.” arXiv, January 13, 2023. http://arxiv.org/abs/2202.05262.

Meng, Kevin, Arnab Sen Sharma, Alex Andonian, Yonatan Belinkov, and David Bau. “Mass-Editing Memory in a Transformer.” arXiv, August 1, 2023. http://arxiv.org/abs/2210.07229.

Olsson, Catherine, Nelson Elhage, Neel Nanda, Nicholas Joseph, Nova DasSarma, Tom Henighan, Ben Mann, et al. “In-Context Learning and Induction Heads.” arXiv, September 23, 2022. https://doi.org/10.48550/arXiv.2209.11895.

Li, Kenneth, Aspen K. Hopkins, David Bau, Fernanda Viégas, Hanspeter Pfister, and Martin Wattenberg. “Emergent World Representations: Exploring a Sequence Model Trained on a Synthetic Task.” arXiv, February 27, 2023. https://doi.org/10.48550/arXiv.2210.13382.

**Questions:**

My questions revolve around addressing the 3 primary weakness I mention above.

1. Besides neuroscience, how does this current work relate to past and ongoing work in mechanistic interpretability? E.g., Meng et al., 2023a, 2023b; Olsson et al., 2022; Li et al., 2023. More specifically, how does the path patching approach used here compare to methods and insights in those papers?

2. Is it possible to include additional figures/visualizations to improve understanding of the paper? More specifically, is it possible to create a clearer visual explanation of the reference-back-2 task or a diagram showing how the query and key vectors function as gating mechanisms? Since the paper aims to interpret mechanisms of transformers, the contribution of the paper hinges on its clarity and presentation.

3. What is the relation of the final result (Fig 5) to curriculum learning?

Minor question: What do the authors hypothesize would happen for multiheaded transformers? What does each attention head do?

---

> ### Author Response · Authors · 2024-11-27
> **Response to Review**
>
> Thank you for the review.
> The novelty of our work does not rely on the fact that we perform path patching on a transformer, but that it’s possible to connect the mechanism a transformer uses to solve a task specifically targeting working memory gating to a mechanism in the brain. Previous work in mechanistic interpretability aims to explain how neural networks solve some task of interest, but have no motivation to design tasks that target this domain. We borrow methods from this literature, but are not claiming to innovate on the methods themselves. Rather, the contribution is that we apply these methods to answer important questions at the intersection of AI and neuroscience. That is, the connection to cognitive neuroscience is not just a nice analogy, it is the central research question in this work.
> We will make clarifications to the text to better explain the task. The key/query analogy to input and output gating is abstract but is proven using some of the corresponding weights - i.e. refer to page 5 (Section 4.1 KEY AND QUERY VECTORS SPECIALIZE FOR INPUT AND OUTPUT GATING). “That is, key vectors representing an Ignore tuple receive very little attention (0.4% of layer 1 attention averaged over test set), whereas those representing a Store tuple receive the bulk of the attention (86.8%).” Suggesting the key vector is similar to input gating due to its differential and systematic treatment of the tokens store and ignore.
> “We find that patching to the query vector in such cases indeed causes the attention to shift from the original stored tuple (74.1% of attention) to the stored tuple that matches the edited register, resulting in a corresponding change in the final same/different judgment.” This suggests the analogy between query and output gating due to the shift of attention to different registers as is related to output gating/pulling out information from working memory.
> Mechanistic training on small scale tasks is slightly different from curriculum learning - this is exemplified with the generalization results comparing tasks of same size, but with different underlying structure. One task allowed the model to learn a mechanistic solution and generalize - but the other task (same registers and symbols) did not. The point here was to train with a proper mechanistic task (and not just any easier task - in fact it is too easy, we conclude that it is not helpful). We will discuss more specific details of curriculum learning and incorporate this into the text.
> We hope to add clarifications to the neuroscience component and reduce the load on the reader trying to understand the main takeaway.
> The path patching methods we use are previously described in the literature(Wang et al., 2022; Goldowsky-Dill et al., 2023). We have not made new discoveries or changes to the methodology.
> The task will be clarified in the text and figure.

---

> > ### Comment · Area_Chair_4d7a · 2024-11-30
> >
> > Dear Reviewer,
> >
> > The authors have provided their responses. Could you please review them and share your feedback?
> >
> > Thank you!

---

> > > ### Comment · Reviewer_2znX · 2024-12-01
> > > **Reply**
> > >
> > > Dear authors,
> > >
> > > Thanks for your reply to my review. In my initial review, I mentioned 3 primary weaknesses with the paper:
> > >
> > > 1. Novelty in relation to prior work in mechanistic interpretability
> > > 2. Clarity of the paper, particularly the description of the task and how it addresses the main problem/domain in this study
> > > 3. The relevance to current findings in relation to curriculum learning in figure 5.
> > >
> > > Though I am sympathetic to the arguments made by the authors, i.e., that 1) the work here addresses central questions in cog neuro, and 2) the work in mechanistic interpretability does not address the domain of topic in this paper (e.g., working memory and gating), it was not immediately clear to me that they have addressed this concern in the manuscript (since no revisions were attached), and if they were, how they might address it.
> > >
> > > Regarding clarity of the task: The authors mention that they will improve and revise the clarity, but do not specifically mention how.
> > >
> > > Regarding the relation to curriculum learning in Figure 5: I am open to the authors argument, i.e., "Mechanistic training on small scale tasks is slightly different from curriculum learning". But I was a bit confused by their explanation -- curriculum learning, as I understand it, encompasses a broad suite of techniques aimed to embed useful representations into models through tasks (not exclusively simpler tasks).
> > >
> > > In sum, because it was not clear how the authors were planning to more specifically address the initial concerns raised beyond mentioning they will revise the manuscript, I will keep my score.

---

### Official Review · Reviewer_CBqC · 2024-11-01

**Soundness:** 1
**Presentation:** 2
**Contribution:** 2
**Rating:** 3
**Confidence:** 3

**Summary:**

In their paper “Transformer mechanisms mimic frontostriatal gating operations when trained on human working memory tasks” the authors train simple transformer-based networks a task used in cognitive neuroscience to study working memory. By using behavioural analysis in combination with some derivative tasks and the path-patching technique from MechInterp they show that transformers can learn to solve the task with Gating operations. They draw conceptual conclusions to human neuroscience where Frontostriatal circuits are believed to implement gating operations as part of working memory as well.

**Strengths:**

I personally enjoyed the authors investigations on identifying how task-dependent gate mechanisms develop through training and how it can be identified through behavioural investigations using thoughtfully constructed task variants. I do not necessarily find their findings surprising, but I acknowledge that having evidence for expected mechanisms is worthwhile scientific work and hence would not let that influence my rating.

**Weaknesses:**

Unfortunately, I do see quite major weaknesses in this paper in its current state. In the following I group my concerns into a block of neuroscience-related concerns and Mechanistic Interpretability concerns.

*Neuroscience-related concerns*

The authors spend quite a bit of their manuscript making conceptual links to the brain’s gating mechanism in working memory but I struggle to see the relevance of their investigations to the brain’s working memory system, for the following reasons:

- Using transformers for working memory investigations: The authors declare themselves that a weakness of their investigations is that transformers have access to the entire input sequence, unlike the brain. In fact, the very idea of working memory is that the immediate past needs to continually compressed into a latent state of activations or rapid connectivity changes (e.g. see Stokes 2015 TICS). I wonder why the authors would not opt to use Mamba-like State Space Models which would seem to be much closer to brain like processes while also allowing for input dependent processing, and potentially slots through the distinct hidden states in Mamba. I see that models need to be abstractions and that one can in principle study working memory with transformers, but I am not aware of any prior investigations showing such links and hence the study here would need to provide such links themselves, which brings me to my next point:
- Comparison to the brain is purely conceptual: Given that the authors focus on the brain so much in both their title, abstract, and introduction I would have expected to see some actual data comparisons to cognitive neuroscience data, but the authors do not seem to provide any. The only data-like comparison they have is that the model struggles more to learn the task with more working memory items which supposedly is similar to humans but that simply seems like a general task difficulty effect and does not link to gating specifically.
- Poorly referenced neuroscience work: The authors heavily rely on the idea of ‘Stripes’ in frontal cortex for working memory for their model to be relevant, as it is about recalling content from distinct memory slots. Recent neuroscientific investigations call the idea of discrete slots implemented through distinct groups of cells into question and instead think of working memory as being implemented by dynamic and distributed population codes (Meyers 2018 JNeurophys; Miller 2013 Dialogues in ClinNeuro). In case that authors make to want a serious link to neuroscience I think they should discuss how their model could be reconciled with the dominant idea of population codes. Also, the 3-4 chunks given as working memory capacity is on the lower end on the scale though I acknowledge that there is some discussion around it. Of course, the classical number to use is 7 +/- 2 (as discussed in the reference the authors give).

*MechInterp-related concerns*

So, the above points are to be taken seriously to make a believable link to neuroscience. Of course, the neuroscience-link is only conceptual, and the main work of the paper actually is in the MechInterp world showing how gating mechanisms develop in trained transformers. As said in the strengths section, that seems to be an interesting analysis though I am not really well qualified to judge how new that finding is. It seems intuitive to me that this should happen and, as the authors mention themselves, models like LSTM were actually constructed specifically with such ideas in mind. If the authors see their main contribution in studying how a gating mechanism develops in transformers, then I would suggest to strongly deemphasize the neuroscience narrative and instead contextualise the research more in the context of existing MechInterp findings. For example, work like [1] from earlier in the year looks at reasoning mechanisms which at least partially seem to rely on mechanisms similar to the ones proposed in this paper and I assume there is additional other work which I am not aware of.

[1] Brinkmann et al, 2024 https://arxiv.org/abs/2402.11917

**Questions:**

Major question:
- Do the authors see their key contribution in understanding transformers of making a strong point that transformers work like the brain? If it is the latter, I would expect a more detailed discussion of neuroscientific theories, at least a comparison of Transformers with models which are more typically considered working memory models with hidden states, and ideally some direct comparison with data from cognitive neuroscience. If it is the former, then I suggest the neuroscience link should be heavily deemphasized to not be misleading about the similarity to the brain. At the same time, I think in that case a reviewer who is well-versed in the MechInterp field should be included in the decision around acceptance. At the very least, the neuroscience content should largely be replaced by an actual overview of related MechInterp papers.

Minor questions:

- Can you add more infos on the actual setup of the data, for example how many timesteps does one trial have? I do not find that information.
- Is there any control to make sure the trials in the validation set are sufficiently different from the training set? Given you use a finite set of symbols with no noise, I wonder whether there are trials identical across datasets? The probability of this is hard to judge given the information about the length of trials seems to be missing. I am sorry if it is there and I just cannot locate it.

---

> ### Author Response · Authors · 2024-11-27
> **Response to review part 1**
>
> Our key contribution is test whether learning difficulties and solutions that are present in humans and biological neural network models (specifically,  credit assignment) can also be seen in Transformers. This is an interesting question because Transformers do not have architectural memory capacity limits, and thus if we see similarities, it suggests that these difficulties stem from more general learning principles. We establish that Transformers can learn gating-like mechanisms that can be leveraged for better generalization accuracy. Transformers are quite distinct from the brain, but we bridge the gap to utilize what we know about the brain to better train Transformers.
>
> The length of each trial has 10 same or different arbitrations: this means initializations for each register and then 10 randomly sampled switches. We have 100,000 unique training examples that are used for each epoch. The validation dataset has unique trials separate from the training - this is validated at the time of data generation.
>
> The reason to use Transformers despite them having access to the entire context (as opposed to mamba or other recurrent models) is deliberate: it isolates the need for WM management as opposed to maintenance.  Indeed, there is growing evidence that what limits human WM capacity is not the demands on maintenance of the number of items that one can store but rather the management of WM, including binding an item to its role, which can be supported by input and output gating in biological models of frontostriatal circuits. Indeed even in these models which do have to maintain information over trials, effective WM capacity is limited by difficulties in this management problem (and learning thereof) and not the number of representations per se (Soni and Frank 2024). In this sense it is stronger to use the Transformer because it has no maintenance capacity limits at all but challenges these computational demands, to isolate the difficulties in management. Our results show that when pressured to do so by task distribution, the Transformer learns input and output gating policies which enable it to manage WM role addressability, and when it does so it can much more rapidly generalize to tasks with higher WM loads.
>
> We don’t include comparisons to real brain data because (i) our focus here is on establishing the computational challenges in Transformers as motivated by those in frontostriatal networks, and (ii) it has also been established over many previous publications across species that the BG and thalamus are involved (and needed)  for gating / controlling access to/from WM  (Cools et al. 2007, McNab et al. 2008, Baier et al. 2010, Astle et al. 2014,  Feldman et al. 2019 Wilhelm et al. 2023), So the contributions here focus on the inductive biases that the brain seems to have (input and output gating mechanisms, which interact with capacity limits)  and how they relate to what Transformers do.

---

> > ### Author Response · Authors · 2024-11-27
> > **Response to review part 2**
> >
> > We indeed should have situated our work in relation to other neuroscience connections to Transformers.  For example, Whitington et al showed how transformers given spatial position encodings relate to models of hippocampus and place/grid cells. Ji-An et al. examine induction heads in transformers and their relation to models of episodic memory for contextual retrieval, and how these can account for position-dependence of memory recall.  While each of these makes its own contribution linking transformers to episodic memory phenomena, they have not addressed this challenge in  reading and writing to distinct addresses in working memory in a fashion that allows them to be retrieved when needed, as motivated by the human reference-back task.
> >
> > Regarding neural data in WM. The idea of dynamic and distributed population codes does not challenge the idea of stripes/slots (or similar). Modern cognitive science models reconcile the differences between slots and resources models, positing hybrid models wherein each slot-like entity can represent multiple items with limited resources (e.g., Nassar et al, 2018), and indeed this has been recently simulated within stripes (Soni & Frank 2024). Early versions of frontostriatal networks suggested computational utility of distributed codes within stripes (but used simpler localist representations for visualization in simpler tasks). Other models in the same family also show how “stripes” need not be anatomically pre-defined but can emerge as clusters within an RNN, but still benefit from gating from BG and thalamus to trigger transitions and support generalization (Calderon et al, 2022).  Computationally, a key function of these models is that a prefrontal population acts as a pointer/role to support variable binding of its content, which also facilitates generalization (O’Reilly & Frank 2006; Kriete et al, 2013; Collins & Frank, 2013) – but whether this is implemented in anatomical pre-defined populations or sub-spaces (and whether the representations therein are fixed point or dynamic attractors) is impertinent to the abstraction. Indeed, Lundqvist et al 2023 recently reviewed data from Miller’s lab in nonhuman primates, and proposed how prefrontal populations perform “spatial computing”,  representing the abstract roles of items separate from their content. Moreover they linked beta dynamics to control processes that modulate access to working memory, and gamma band dynamics to the information content itself. Note that gating decisions by basal ganglia and thalamus themselves will trigger transient dynamics in the cortex which are often studied in the beta band. The authors found support for this spatial computing notion across multiple datasets, which is largely consistent with that predicted by frontostriatal gating.

---

> > > ### Comment · Reviewer_CBqC · 2024-11-28
> > >
> > > Dear authors,
> > >
> > > thank you so much for taking the time to respond to my review. Can I quickly clarify whether the responses you provided above are reflected in changes to the manuscript's text? If yes, could you point me to where these were clarified in the text? Thank you for your efforts!

---

### Official Review · Reviewer_dX8K · 2024-11-03

**Soundness:** 3
**Presentation:** 3
**Contribution:** 2
**Rating:** 5
**Confidence:** 4

**Summary:**

This paper explores how Transformers, when trained on human working memory tasks, develop mechanisms that relate to frontostriatal gating operations observed in human cognition. The study focuses on understanding whether Transformers can solve working memory tasks using self-attention to mimic input and output gating, which are key to human working memory function. By training a small, attention-only Transformer model on tasks requiring selective memory gating, the authors find that certain task conditions lead to the emergence of gating-like behaviors in the model. The results suggest that these emergent mechanisms enhance the model’s generalization and task performance, potentially bridging cognitive neuroscience and artificial intelligence.

**Strengths:**

Originality: The study’s focus on emergent mechanisms in Transformer models trained on working memory tasks is novel. The observation that Transformers struggle with the three-register task is surprising. This research adds a unique perspective to both AI and cognitive science by applying Transformers to a working memory framework typically explored in neuroscience.

Quality: The study employs a simplified, attention-only Transformer, enhancing interpretability and enabling the identification of specific mechanisms within the model. The identification of Transformer mechanisms sheds light on how these models might solve working memory tasks and how specific task demands can trigger such emergent functionality. Control tasks are designed to isolate the conditions necessary for mechanistic emergence, strengthening the quality of the findings.

Clarity: The analysis and results are clearly presented, making it accessible for broad readers.

Significance: Understanding how Transformers solve tasks requiring working memory could substantially impact Transformers interpretability and development. By drawing parallels between Transformer mechanisms and human frontostriatal gating, this paper contributes valuable insights for interdisciplinary research. The study’s findings bridge AI with cognitive neuroscience, offering a model that informs both fields and fosters further exploration into the intersection of biological and artificial intelligent systems.

**Weaknesses:**

•	While the use of a small Transformer model with few heads allows for easier interpretability, the study does not fully analyze or demonstrate the distinct functions of each attention head in each layer. For instance, why exactly are two heads required per layer, and what distinct functions do each head serve? More detailed analyses of each head’s role (e.g., see below) would enhance the interpretability of the model’s mechanisms, and thus the impact of the work.

•	A comparative diagram for understanding “working memory” mechanisms between Transformers and RNNs—often used in biological working memory research—would provide useful context for readers.

•	In Figure 2, while path patching examples are shown, it is implied rather than explicated stated which layer and head are depicted, making it somewhat unclear which specific head is involved.

•	For brevity, I use x^0_i, x^1_i, x^2_i to specify the residual stream at tuple i before the first layer's output, after the first layer's output, and after the second layer's output, respectively. The layer 1 head appears to bind the components (Ins_i, Reg_i, Sym_i) into a tuple and write it to the residual stream x^1_i of Sym_i. However, it is not fully explained if the layer 1 head writes the Store or Ignore Ins_i to different directions within the residual stream. If so, it may function as part of the "input gating" mechanism (suggesting that the Store direction in layer 1 head’s output, but not Ignore direction, is similar to a "working memory" space). Further, do the two heads function similarly or differently?

•	The functions of the layer 2 head, especially how it compares Reg_i and Reg_j (based on Ins_j=Store), are not fully elucidated. A detailed explanation of how the layer 2 head compares the current Sym_i and a prior Sym_j to output same/different answers would clarify its role in “output gating”. Do the two heads function similarly or differently? The study could use mechanistic interpretability tools, such as analyzing QK and OV circuits, or examining the geometric alignment of vectors in key, query, and value subspaces to address these gaps.

•	While transformers can be trained on working memory tasks, they inherently differ from biological systems as they have access to all previous residual stream positions. The most suitable counterpart of “working memory” might be x^1_i, which layer 1 head writes into and layer 2 head reads from. However, x^1_i is different for each position i, different from a true hidden-state-like "working memory". This divergence raises significant concerns about labeling the mechanisms as "input gating" and "output gating" or referring to Transformer mechanisms as "working memory". Using different terms across the manuscript, or discussing the limitations of these analogies would make the paper’s framing more precise.

•	The introduction lacks a review of previous work on comparing Transformers to brain/neuroscience (to name a few, https://arxiv.org/abs/2112.04035, https://arxiv.org/abs/2405.14992), which could more accurately depict the gap and contextualize the paper’s contributions.

**Questions:**

1.	Given that the layer 2 head primarily computes functions linearly (Elhage et al 2021), has the author considered potential failure modes? For example, layer 2 head attends more to the Store tuples than Ignore tuples (based on layer 1 head's output?), and attend more to the most recent Store tuples than more distant Store tuples (based on position embedding?). Consider the scenario of a Store tuple followed by many Ignore tuples—could a recent Ignore tuple receive higher attention than distant Store tuples if position embedding outweighs the Store feature (due to linearity in key-query operation)? Such a failure mode would be significant, as RNNs, unlike Transformers, are less susceptible to this (assumed learned attractor dynamics within the hidden state representation).

2.	Could the authors consider using mechanistic interpretability tools, such as QK and OV circuit analysis, or representational geometry mapping of key, query, and value spaces? These analyses help further clarify how the mechanisms are implemented within each head and layer.

3. For other questions please see weaknesses.

---

> ### Author Response · Authors · 2024-11-27
> **Response to Review**
>
> Thank you for your review.
>
> Separate head analysis would be interesting and better elucidate the gating that is occurring. In this paper, we aimed to show that attentional head mechanisms could look similar to fronto-striatal gating. We sought the most simple analysis that would support this claim, acknowledging that there is room for even more fine-grained circuit analysis.
>
> A diagram for the Transformers’ working memory was not provided because the Transformer has no inherent memory component. As noted by the reviewer, Transformers are inherently different from biological systems because of this reason. The main reason for using Transformers to test on working memory tasks was because they have “unlimited” working memory. Prior work (Braver et al. 2008,  Soni and Frank 2024) in biologically based systems show that working memory capacity limitations may emerge partly due to the learning problem: learning how to utilize working memory resources. We wanted to observe if this learning problem was also present in Transformers (despite having an “unlimited working memory”). We showed this using the difficulty the model has with the 3 register task.
>
>
> Our path patching experiments extract embeddings from both heads of the first layer and send to both layers of the second layer.
>
> Multiple Ignore tuples are possible, and when the model has correctly identified the meaning of store and ignore, this is not a concern since the context window of the transformer is unlimited. This becomes more of a problem in biological or biologically-inspired systems.
>
> Path patching works by localizing effects to either the QK or OV circuits, but is the reviewer referring to analysis of these circuits through the embedding matrices (W_u*QK*W_e) like what is performed in Elhage et al., 2021? If the reviewer could explain what analysis they are interested in seeing and what specifically they think this analysis would provide, we could better respond to the request and discuss feasibility and associated tradeoffs. Generally speaking, we agree that a better characterization of individual head behaviors would help communicate how exactly the gating is implemented, though this is a sufficiently open-ended direction to reasonably be left for future work.

---

> > ### Comment · Area_Chair_4d7a · 2024-11-30
> >
> > Dear Reviewer,
> >
> > The authors have provided their responses. Could you please review them and share your feedback?
> >
> > Thank you!

---

> > ### Comment · Reviewer_dX8K · 2024-11-30
> >
> > Thank you for your thoughtful reply. I still believe that for a clear comparison between attentional head mechanisms and frontostriatal gating, a more detailed interpretability analysis is essential to fully support the claim. Such an analysis would provide a clearer understanding of how the first- and second-layer heads function and contribute to input and output gating. For instance, explicitly delineating the role of the first-layer head in "input gating" would enable a more precise comparison with biological mechanisms.
> >
> > Regarding the potential failure mode:
> > The layer 2 head appears to prioritize attention to Store tuples over Ignore tuples (based on Store/Ignore information). Additionally, it seems to favor more recent Store tuples over more distant ones (informed by position embeddings). However, since Store/Ignore information and position embedding are combined nearly linearly within the layer 2 head (Elhage et al., 2021), it is possible for the position embedding to outweigh the Store/Ignore information. Based on this, my hypothesis is that the output logit difference between the correct and incorrect answers decreases as the number of consecutive Ignore tuples increases. This failure mode, if present, could significantly differentiate Transformers from RNNs, which might rely on attractor dynamics within hidden state representations to mitigate such failure mode.

---

### Meta-Review · Area_Chair_4d7a · 2024-12-13

**Metareview:**

This work shows that Transformers, when trained on tasks requiring working memory gating, develop input and output gating mechanisms that resemble those in human frontostriatal systems.

The paper is generally well-written and aims to bridge the gap between gating mechanisms in AI Transformers and frontostriatal systems in biological brains. The approach provides valuable insights into the interpretability of the self-attention mechanism in Transformers.

However, all three reviewers rated the paper below the acceptance threshold due to several unresolved issues, even after the rebuttal:

Lack of comparisons with other AI architectures, such as Memba and RNNs, and insufficient justification for selecting Transformers as the focus.
Insufficient references and comparisons to existing works linking Transformers to neuroscience.
Absence of behavioral or neuronal-level comparisons with neuroscience data from biological brains.

The authors are encouraged to address these concerns and include the suggestions in future submissions.

**Additional Comments On Reviewer Discussion:**

All three reviewers rated the paper below the acceptance threshold due to several unresolved issues, even after the rebuttal:

Lack of comparisons with other AI architectures, such as Memba and RNNs, and insufficient justification for selecting Transformers as the focus.
Insufficient references and comparisons to existing works linking Transformers to neuroscience.
Absence of behavioral or neuronal-level comparisons with neuroscience data from biological brains.

---

### Decision · Program_Chairs · 2025-01-22

Reject